# Asynchronicity of endemic and emerging mosquito-borne disease outbreaks in the Dominican Republic

Mary E. Petrone [1,4✉], Rebecca Earnest[1,4], José Lourenço [2], Moritz U. G. Kraemer [2], Robert Paulino-Ramirez[3], Nathan D. Grubaugh [1,4] & Leandro Tapia [3,4✉]

Mosquito-borne viruses threaten the Caribbean due to the region's tropical climate and seasonal reception of international tourists. Outbreaks of chikungunya and Zika have demonstrated the rapidity with which these viruses can spread. Concurrently, dengue fever cases have climbed over the past decade. Sustainable disease control measures are urgently needed to quell virus transmission and prevent future outbreaks. Here, to improve upon current control methods, we analyze temporal and spatial patterns of chikungunya, Zika, and dengue outbreaks reported in the Dominican Republic between 2012 and 2018. The viruses that cause these outbreaks are transmitted by *Aedes* mosquitoes, which are sensitive to seasonal climatological variability. We evaluate whether climate and the spatio-temporal dynamics of dengue outbreaks could explain patterns of emerging disease outbreaks. We find that emerging disease outbreaks were robust to the climatological and spatio-temporal constraints defining seasonal dengue outbreak dynamics, indicating that constant surveillance is required to prevent future health crises.

---

[1] Department of Epidemiology of Microbial Diseases, Yale School of Public Health, New Haven, CT 06510, USA. [2] Department of Zoology, University of Oxford, Oxford, United Kingdom. [3] Instituto de Medicina Tropical & Salud Global, Universidad Iberoamericana, Santo Domingo, Dominican Republic. [4] These authors contributed equally: Mary E. Petrone, Rebecca Earnest, Nathan D. Grubaugh, Leandro Tapia. ✉email: mary.petrone@yale.edu; l.tapia@prof.unibe.edu.do

Emerging and endemic mosquito-borne viruses are a constant public health concern in the Caribbean[1–3]. This region is especially vulnerable to the spread of the former owing to its tropical climate and large tourism industry, which attracts visitors from all across the globe. The recent outbreaks of chikungunya[4–6] and Zika[7–9] in 2014 and 2016, respectively, demonstrated that viruses once associated with mild illness can re-emerge and cause devastating health outcomes. Reports that Mayaro virus, which has already been detected in the Caribbean, can be transmitted by the urbanized mosquito species *Aedes* indicate that future outbreaks of emerging mosquito-borne diseases may be on the horizon[10–14]. In addition to this threat, dengue virus is endemic to many Caribbean countries and territories and has caused outbreaks with increased frequency over the past decade. Large outbreaks, which began in 2019 but have continued through 2020[15], extend an alarming trend of a rising number of dengue cases reported annually in the Americas in recent years[16]. However, the danger of these viruses and the diseases they cause lie not only in their debilitating and sometimes life-threatening symptoms. The unpredictability of when and where a new outbreak will occur precludes preparedness. Outbreak response strategies are inherently reactionary and, in their transience, disrupt public health systems when new initiatives are introduced and subsequently phased out. This is especially problematic in resource-limited settings where the strategic allocation of resources should be prioritized to maximize the impact of disease control efforts. New approaches centered around sustainable, long-term surveillance are needed to curtail the potential for future public health crises in the Caribbean.

One such approach is the use of climate data to evaluate the risk of viral spread. The transmission of mosquito-borne viruses, both emerging and endemic, should adhere to similar temporal and spatial patterns such that the dynamics of past outbreaks can inform those of future outbreaks. Many flaviviruses, like dengue and Zika viruses, and alphaviruses, like chikungunya virus, are transmitted by *Aedes* mosquitoes, which are sensitive to climatological variability[17–19]. Temporal concordance between seasonal weather patterns and transmission of yellow fever virus (flavivirus) has been documented on the African continent[20]. Similar associations have been reported for dengue fever incidence in Hanoi, Vietnam, and Myanmar[21,22] and for chikungunya and Zika incidence in the Americas[17]. However, there is evidence to suggest that this relationship does not extend to the transmission dynamics of viruses during their first year of introduction into a new population[23,24]. Therefore, we considered whether, whereas climate may be a useful indicator for future endemic virus outbreaks, other factors including population size, demographics, and the timing of introduction should be considered when developing strategies to prevent future emerging disease outbreaks.

To answer these questions in the context of the Caribbean, we analyzed dengue, chikungunya, and Zika cases reported daily in the Dominican Republic between 2012 and 2018. We found that emerging disease outbreaks (chikungunya and Zika) occurred earlier in the year than dengue outbreaks, and the timing and location of introductions of emerging viruses impacted when and where corresponding outbreaks occurred. Moreover, the spread of chikungunya and Zika viruses was tolerant to suboptimal climates for transmission by *Aedes* mosquitoes, likely due to the large size of the susceptible human population. Predicted mosquito abundance was similarly uninformative for the spatial distribution of emerging disease attack rates and force of infection. Moreover, provincial-level dengue attack rates were consistent between dengue outbreaks, but they did not correspond to local attack rates of chikungunya and Zika. Taken together, we demonstrate that dengue virus may not always be an appropriate model to prepare for future emerging mosquito-borne disease outbreaks. Instead, a sustainable and long-term mosquito-borne disease surveillance system is needed to facilitate proactive responses to emerging threats and to track the continued spread of known diseases including dengue, chikungunya, and Zika. We specifically propose the use of two indicators, the incidence of febrile illness cases and dengue case fatality rates (CFRs), to monitor surveillance performance and identify potential emerging threats.

## Results

**Multiple endemic and emerging mosquito-borne virus outbreaks in the Dominican Republic.** Between 2012 and 2018, the Dominican Republic, a country that shares the Caribbean island La Hispaniola with Haiti (Fig. 1a), experienced five disease outbreaks caused by mosquito-borne viruses (Fig. 1b, c). We divided this period into six seasons (Seasons 1–6), each beginning in April of every year between 2012 and 2018, coincident with the start of the rainy season. Three of the five outbreaks were caused by dengue virus (Seasons 1, 2, and 4; Fig. 1b, c). Sufficient serosurvey data for the Dominican Republic are not available to determine the predominant dengue virus serotype(s) during each outbreak, but the Pan-American Health Organization reports that serotypes 1, 2, and 4 were prevalent in the Caribbean between 2012 and 2014, and serotypes 2, 3, and 4 were circulating in the region in 2015[25]. The other two outbreaks were caused by emerging viruses, chikungunya and Zika (Seasons 3 and 5, respectively; Fig. 1b, c)[26]. Those outbreaks were the first and only to be reported of either disease in the country, and the number of reported cases in the Caribbean as a whole plummeted in subsequent years[27,28]. After the season dominated by Zika virus (Season 5), only a small number of dengue cases were diagnosed (Season 6).

Clinical and demographic characteristics differed between reported dengue, chikungunya, and Zika cases. The majority of dengue cases experienced fever (98.3%), were hospitalized for their condition (93.7%), and were between the ages of 0 and 15 (62.3%), suggesting immunity in the adult population (Table 1). Patients diagnosed with chikungunya and Zika were older and predominantly female. Rash and/or arthralgia in conjunction with fever are typical clinical manifestations of chikungunya and Zika infections[29–34]. Although lower than rates reported elsewhere, arthralgia rates among chikungunya cases in our data set were consistent with previously documented rates among individuals diagnosed with chikungunya in the Dominican Republic[34].

**Chikungunya and Zika outbreak dynamics did not conform to seasonal dengue patterns.** Implementing effective disease prevention and control measures requires knowledge of when their etiological viruses are most likely to emerge, re-emerge, and spread[35]. Dengue, chikungunya, and Zika are caused by RNA viruses transmitted by the mosquito vectors *Aedes aegypti* and *Aedes albopictus*. The abundance and capacity of these vectors are sensitive to climatological factors including temperature, rainfall, and humidity[19,36], such that transmission of all three viruses has been shown to fluctuate in kind[17,20–23]. We therefore hypothesized that the timing of the five outbreaks should concord with seasonal weather patterns.

We found that the emerging disease outbreaks (chikungunya [Season 3] and Zika [Season 5]) occurred earlier in the year compared to endemic outbreaks (dengue). The chikungunya and Zika outbreaks peaked 15 and 26 weeks earlier, respectively, than the averaged dengue peak (epidemiological week 41; Fig. 1b). While both chikungunya and dengue cases began to rise around

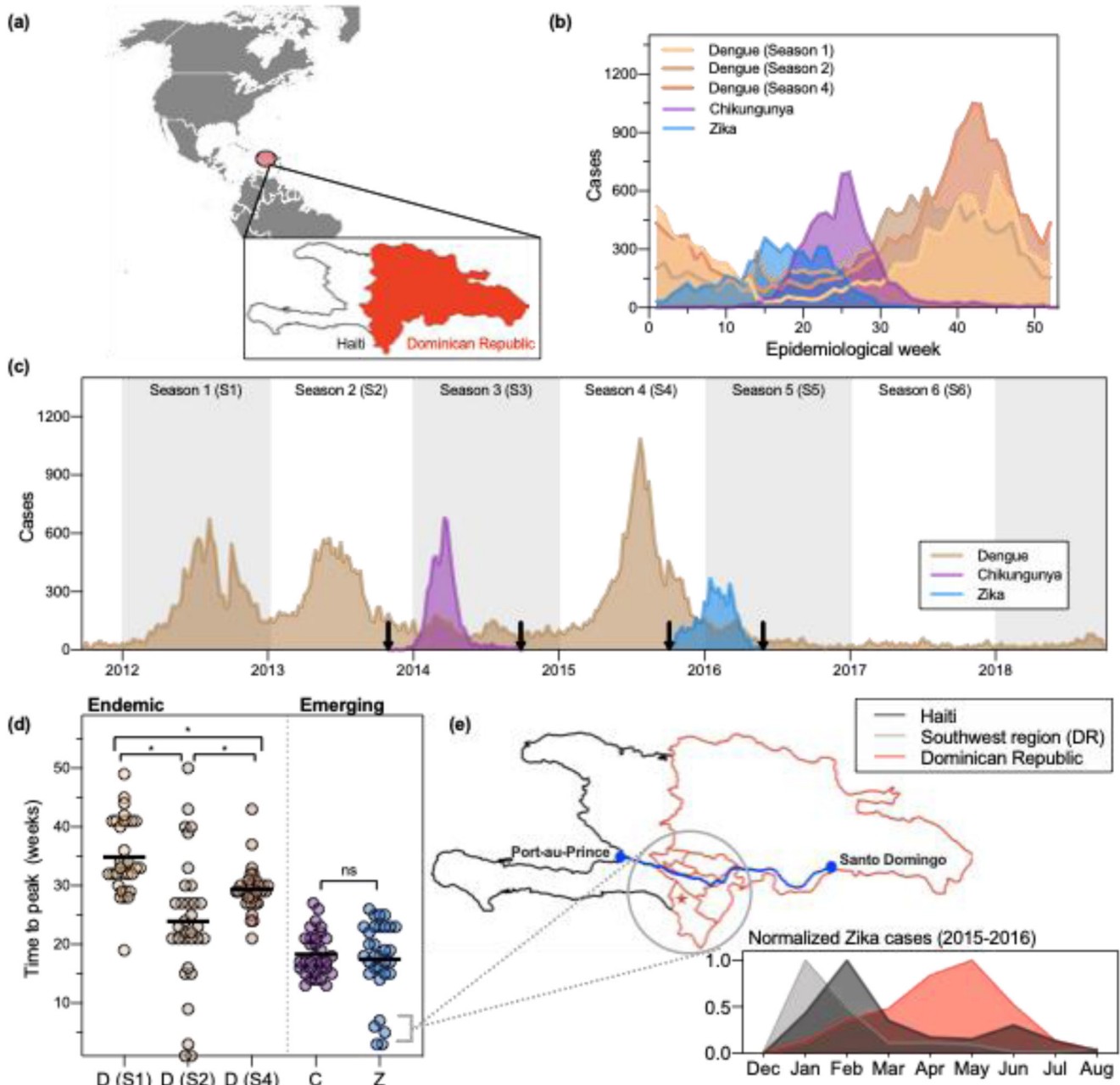

**Fig. 1 Two outbreaks of emerging disease, chikungunya and Zika, and three endemic dengue outbreaks were reported between 2012 and 2018 in the Dominican Republic. a** The Dominican Republic is a country in the Caribbean that shares the island of La Hispaniola with Haiti. Map source (gray): free vector maps. **b** Cases in the Dominican Republic reported by the Ministry of Health during each outbreak per epidemiological week for five seasons. **c** Weekly reported cases from 2012 to 2018. Arrows indicate when the first and last cases were reported for emerging pathogens. **d** Number of weeks elapsed between the first case reported nationally and peak cases reported in 32 provinces (each data point represents a different province). The chikungunya outbreak began during epidemiological week (EW) 6, the Zika outbreak during EW 1, and the three dengue outbreaks during EW 14. The mean time to peak for each outbreak were compared using ordinary one-way ANOVA. We adjusted for multiple comparisons using Tukey's multiple comparisons test. The qualitative statistical result of the analysis did not change when the five early Zika provinces were excluded. Symbols: "ns" indicates $p$ value > 0.05, *indicates $p \leq 0.05$, horizontal bars indicate the median time to peak for each outbreak. D(S1) vs D(S2): $p < 0.0001$; D(S1) vs D(S4): $p = 0.0155$; D(S2) vs D(S4): $p = 0.0173$; C vs Z: $p = 0.9885$. **e** Geographic and temporal comparisons of Zika cases reported by four provinces in the southwestern region of the Dominican Republic. A major roadway connects the Haitian capital Port-au-Prince to the southwestern region of the Dominican Republic (blue line, map) and Santo Domingo, the national capital. A large binational market is located in the province Pedernales (star, map). Similar to Haiti, provinces in the southwest experienced an earlier outbreak of Zika compared with the remaining 28 provinces (inset).

epidemiological week 15, a rise in Zika cases was observed during the first weeks of the year.

Because the implementation of national-level reporting of chikungunya and Zika cases influenced when these outbreaks were detected (Fig. 1c, arrows), we calculated the number of

weeks elapsed between the first reported case of each outbreak and the peak number of cases within provinces (Fig. 1d). We assumed that climatological factors did not vary widely between provinces during a given season. Our reasoning followed that the rate of viral transmission is limited by the extrinsic incubation

**Table 1 Epidemiological and clinical characteristics of cases.**

| Season and disease | Year | Number of cases | Median age of cases (IQR) | % Hospitalized | % Female | % Reported fever | % Reported rash | % Reported arthralgia |
|---|---|---|---|---|---|---|---|---|
| Season 1 (dengue)[a] | 2012–2013 | 13,666 | 11 (11) | 94.3 | 44.0 | 98.7 | 15.0 | 31.0 |
| Season 2 (dengue)[a] | 2013–2014 | 15,079 | 11 (11) | 92.9 | 46.3 | 97.4 | 5.6 | 16.4 |
| Season 3 (dengue) | 2014–2015 | 5236 | 13 (18) | 88.7 | 45.0 | 96.1 | 8.0 | 23.2 |
| Season 4 (dengue)[a] | 2015–2016 | 19,619 | 12 (13) | 93.9 | 46.0 | 98.9 | 4.0 | 17.6 |
| Season 5 (dengue) | 2016–2017 | 3093 | 15 (25) | 85.5 | 42.4 | 98.1 | 7.1 | 24.9 |
| Season 6 (dengue) | 2017–2018 | 1437 | 18 (24) | 87.3[b] | 38.5 | 98.3 | 4.0 | 27.4 |
| Chikungunya | 2014 | 6461 | 28 (32) | 32.0 | 60.8 | 89.5 | 26.0 | 63.0 |
| Zika | 2016 | 5161 | 30 (21) | 16.1 | 73.9 | 74.2 | 70.7 | 8.5 |

IQR interquartile range.
[a]Outbreak reported.
[b]The type of medical attention received by 22.9% of the reported cases during this period was not reported and was assumed to be hospital admissions.

period (EIP) of the virus in the mosquito vector[37], and the EIP for *Aedes* mosquitoes is mainly influenced by temperature[38–40]. Therefore, because we did not expect transmission rates to vary widely between provinces, provincial outbreaks that peaked soon after the reporting system was implemented would indicate that transmission in those provinces preceded the establishment of this system.

Our analysis identified one such instance. The majority of provincial chikungunya and Zika outbreaks peaked within 12 and 25 weeks, and the mean time to peak across provinces did not significantly differ (Fig. 1d, e). This was in contrast to the substantial heterogeneity of the timing of the three dengue outbreaks. The lack of uniformity across these outbreaks may have been due to a slower rate of spread between regions because of a pre-existing and spatially heterogeneous herd immunity landscape. Five provinces, four of which are located in the southwestern part of the country, reported peak numbers of Zika cases in January and February of 2016 (Fig. 1d). For the reasons stated above and because Zika virus had been circulating in the Americas for at least 2 years before it was reported in the Dominican Republic[7,41], this observation suggested that Zika virus was already circulating in those provinces before the national reporting system was implemented at the beginning of January. We hypothesized that early Zika virus transmission could have occurred in the southwestern region because of the region's proximity to Haiti. In particular, this region shares a border with Haiti, is connected to the Haitian capital Port-au-Prince via a major roadway, and is home to a large binational market (Fig. 1e, *map, star*). A temporal comparison of the Zika outbreaks reported in this region, the rest of the country, and Haiti revealed that the regional outbreak was more consistent with the 2016 Zika outbreak reported in Haiti (Fig. 1e, inset). Moreover, the proportion of dengue cases reporting a rash was larger and peaked earlier in the southwestern region Enriquillo in 2016 compared with the rest of the country (Supplementary Fig. 1). This suggests that some Zika cases may have been misclassified as dengue at the beginning of the outbreak. These findings indicate that the Dominican Republic experienced two geographically and temporally distinct Zika outbreaks. From reported case counts alone we cannot conclude whether Zika virus was introduced into the Dominican Republic from Haiti or vice versa; however, it is clear that binational coordination is an essential component of local mosquito-borne disease control because the viruses these vectors transmit do not recognize international borders.

**Initial outbreaks of emerging mosquito-borne diseases can occur during periods of suboptimal climatological conditions.** Having observed that the emerging disease outbreaks peaked earlier in the year than the three dengue outbreaks (Fig. 1), we considered two possible explanations. (1) Either climate patterns differed between seasons and the emerging outbreaks preceded those of dengue due to seasonal stochasticity, or (2) the spread of emerging viruses was less sensitive to climatological factors compared with dengue virus. By analyzing daily climatological data collected over the duration of our study period, we found support for the latter (Fig. 2).

To investigate the relationship between climate and case incidence, we used temperature and humidity time series data to estimate the mosquito-borne transmission potential throughout our study period (Fig. 2). For this analysis, we used *Index P*, a metric that is calculated by incorporating climate and entomological data into a Bayesian framework to estimate the transmission potential of individual female mosquitoes[17]. We calculated transmission potential for each week of our study

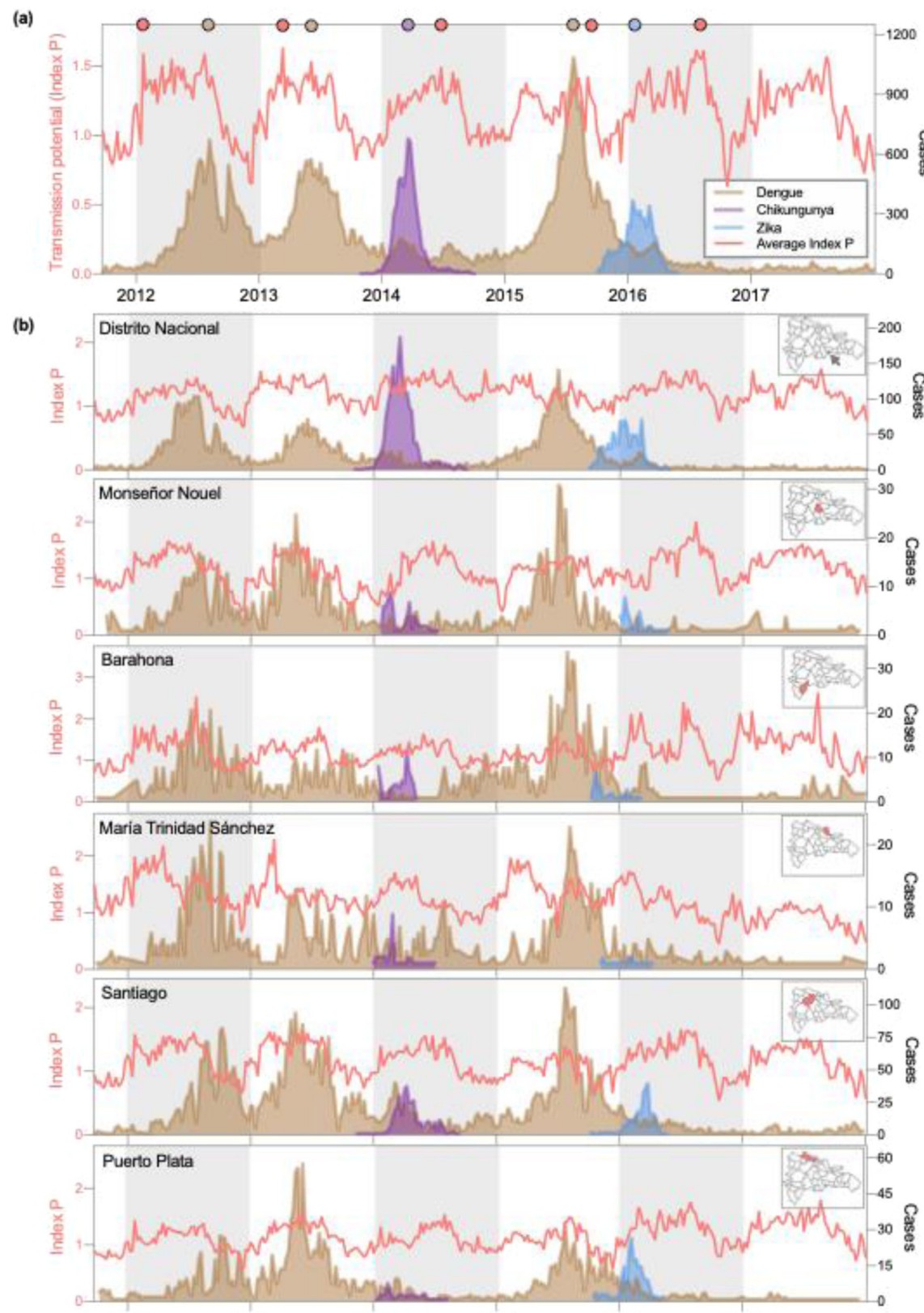

**Fig. 2 Emerging disease outbreaks occurred during periods of suboptimal climate. a** Average weekly transmission potential (*Index P*) and nationally reported cases. *Index P* describes the reproductive potential of an individual female *Aedes* mosquito, meaning its absolute value should be interpreted biologically rather than epidemiologically. Relative fluctuations in *Index P* reflect seasonal changes in the expected rate of mosquito-borne virus transmission. The peaks of each curve during each season are denoted as circles on the top *x* axis. **b** *Index P* and reported cases for six provinces. Temperature and humidity data were retrieved from the National Meteorology Office (ONAMET) database (saip.gob.do) for Distrito Nacional, Barahona, Puerto Plata, María Trinidad Sánchez, and Santiago. Hourly climatological data for Monseñor Nouel were obtained from Open Weather Map (openweathermap.org). Reported cases of dengue, chikungunya, and Zika were organized by province based on the individual's place of residence.

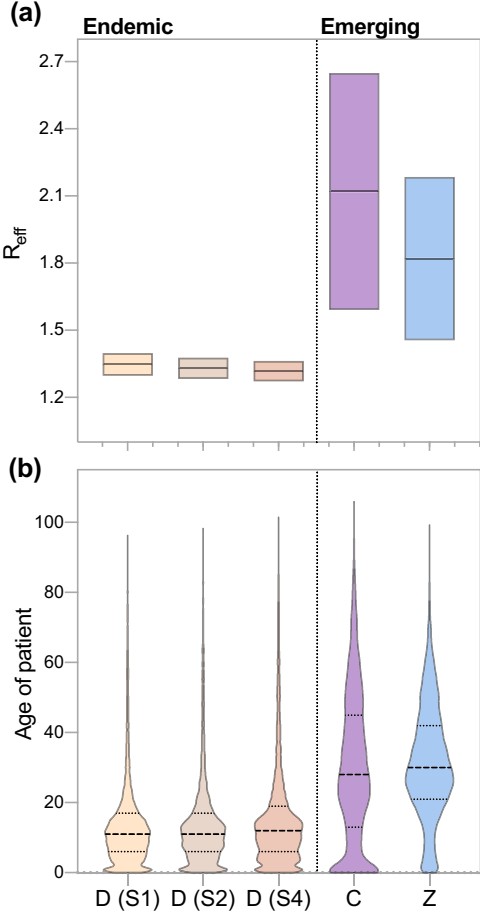

**Fig. 3 The size of the susceptible population was larger for emerging diseases than dengue. a** The effective reproduction number ($R_{eff}$) of chikungunya and Zika far exceeded that of dengue. $R_{eff}$ was calculated using the growth rate of each outbreak[87] and estimates of the incubation period and serial interval of each disease (Supplementary Table 2). **b** The median age of reported cases during emerging disease outbreaks significantly exceeded that of the three dengue outbreaks. The median age group of the national population is 20–25 as calculated from population data reported in the 2010 census. Horizontal lines (dotted) indicate the median age for each group of cases.

period using data reported in 6 of 32 provinces. We selected these six provinces based on their representativeness of the country's sub-climates and data availability. On average, transmission potential fluctuated seasonally, rising between April and November of each year, coincident with the rainy season, and fell shortly thereafter (Fig. 2). When we compared the temporal dynamics of transmission potential to reported disease incidence by visual inspection, we found that the number of emerging disease cases reported weekly peaked before transmission potential had reached a seasonal maximum for both outbreaks (chikungunya and Zika), whereas the number of dengue cases reported weekly peaked after this point for two of the three dengue outbreaks (Fig. 2a). We measured the number of weeks elapsed between outbreak cases and Index P during each season and found that, on average, dengue cases peaked 17 weeks after Index P. In contrast, chikungunya cases peaked 14 weeks and Zika cases peaked 27 weeks earlier than Index P (Fig. 2a, circles). Similar climatological patterns persisted on the provincial level (Fig. 2b). Some provinces experienced clear seasonal fluctuations in transmission potential, whereas seasonal peaks and troughs were

less defined in Distrito Nacional and María Trinidad Sánchez, which have low to moderately humid sub-climates. Despite this variation, the timing of provincial outbreaks conformed to the trend observed with visual examination on the national level: dengue outbreaks peaked after a period characterized by high transmission potential, just as transmission potential was beginning to wane; in contrast emerging disease outbreaks occurred concurrently with increasing transmission potential. Moreover, we cannot attribute climatic anomalies to the decline in reported dengue cases in Seasons 3, 5, and 6.

**The sizes of the susceptible human population influenced the speed at which emerging viruses spread.** Variability in seasonal weather patterns and vectorial capacity did not account for differences in the timing of emerging disease outbreaks (Fig. 2). Rather, we hypothesized that the patterns observed for dengue were likely to have been influenced by a pre-existing and spatially heterogeneous herd immunity landscape. This would mandate that transmission potential must remain high for an extended period before epidemic growth is achieved each season.

To test the hypothesis that local susceptibility influenced the timing of the outbreaks, we examined the relationship between the size of the susceptible human population and each outbreak's doubling time (Fig. 3). We estimated that the median estimated effective reproduction number ($R_{eff}$) for the three dengue outbreaks was between 1.3 and 1.4, whereas those of the chikungunya and Zika outbreaks ranged from 1.45 to 2.65 (Fig. 3a). These findings are consistent with the epidemiology of the three diseases and previous estimates of $R_{eff}$ for chikungunya and Zika[26,42,43]. Published estimates of the basic reproduction number ($R_0$) for dengue, equivalent to $R_{eff}$ in a wholly susceptible population, vary widely (0.97–65) owing to the parameter's sensitivity to the size of the local susceptible population[44]. We observed a slightly longer doubling time (smaller $R_{eff}$) during the third dengue outbreak, consistent with increasing levels of herd immunity from the two previous outbreaks. In our dataset, reported dengue cases were primarily in the 0–15 age group, indicating older age groups had high levels of pre-existing immunity and were not susceptible to disease (Table 1; Fig. 3b). In contrast, we speculated that the entire population was susceptible to chikungunya and Zika, allowing these viruses to spread despite suboptimal weather conditions, facilitating an earlier outbreak peak, and affecting a much wider range of ages (Table 1; Fig. 3b).

**Spatial distribution of dengue attack rates is a poor indicator for initial outbreaks of emerging disease.** Given that the timing of emerging disease outbreaks (chikungunya and Zika) did not conform to that of dengue outbreaks (Figs. 1, 2), we suspected that the spatial distribution of dengue cases would be equally uninformative for chikungunya and Zika outbreaks. Specifically, we hypothesized that the relative burden of dengue within provinces during individual outbreaks would correlate with the burden of dengue during subsequent outbreaks but not with the relative burden of emerging disease cases.

To address this question, we measured the age- and sex-adjusted attack rates by province for each of the outbreaks. We found that the attack rates for individual provinces across outbreaks were well-correlated between dengue outbreaks and between chikungunya and Zika outbreaks, but not across the endemic and emerging viruses (Fig. 4a, b).

Next, we investigated the role of climate and land-use in perpetuating this trend, reasoning that larger mosquito populations would facilitate higher attack rates[36,45]. To this end, we compared an *Aedes aegypti* suitability score (AaS; Fig. 4c), a

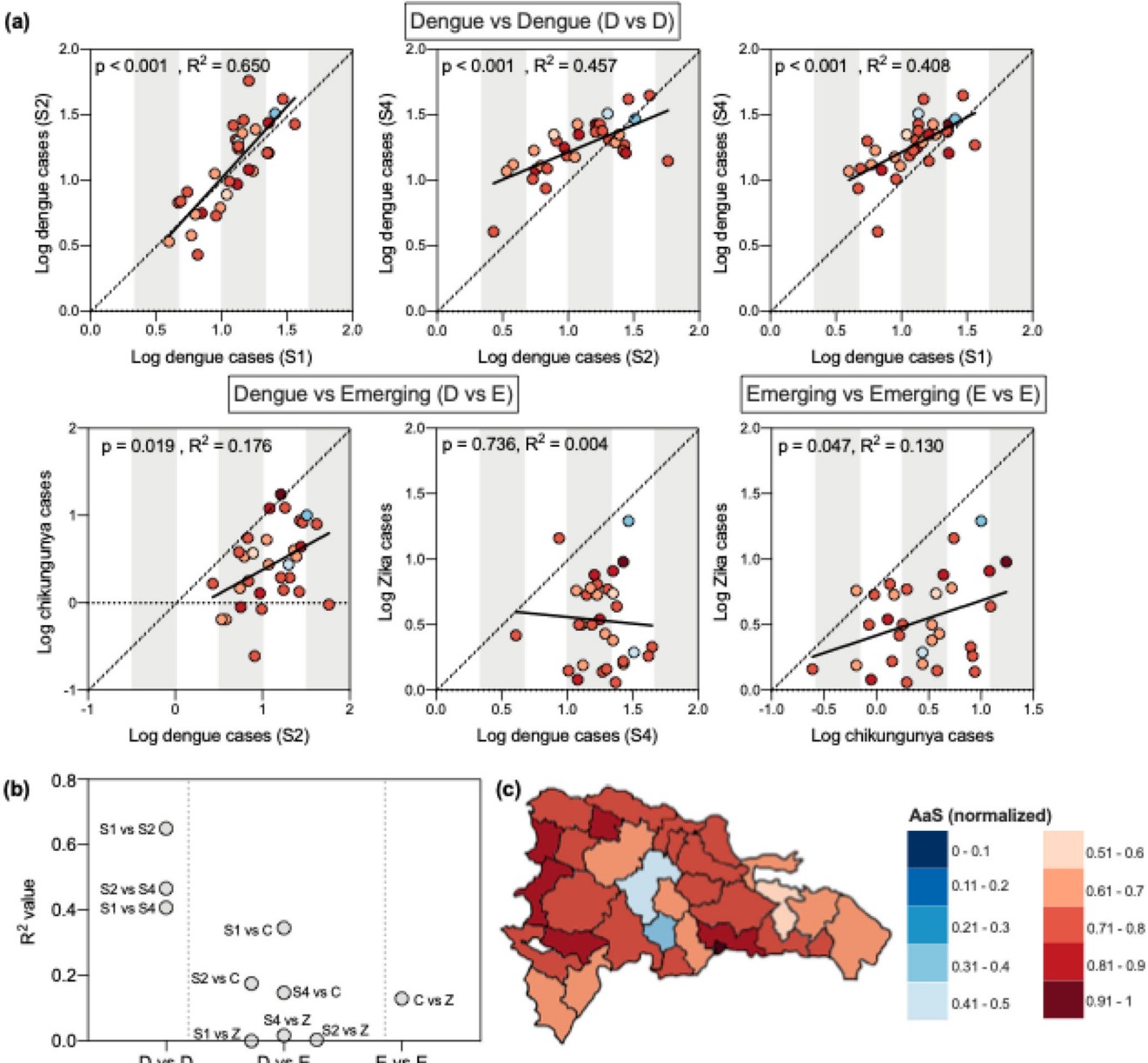

**Fig. 4 The spatial distribution of dengue cases is not well correlated with that of emerging disease cases or with estimated mosquito abundance.**
**a** Population-, age-, and sex-adjusted attack rates were significantly correlated within provinces across dengue outbreaks and between Zika and chikungunya outbreaks. Simple linear regression was used to measure the correlation between the log number of cases reported during each outbreak by province. One province, Elías Piña, was excluded because of assumed substantial under-reporting: the province reported one dengue case during Season 1 and 1 chikungunya case in a population of 63,250. **b** $R^2$ values for dengue vs dengue (D vs D), dengue vs emerging (D vs E), and emerging vs emerging (E vs E) calculated by linear regression. **c** Mosquito abundance (AaS) was normalized across provinces.

metric that incorporates ecological variables not included in our transmission potential estimates such as vegetation levels and precipitation, to population-adjusted attack rates of disease (Supplementary Fig. 2, Fig. 4a). Also unlike our transmission potential estimates, AaS is an aggregated estimate of suitability per month in any given year. Interestingly, AaS was not well correlated with the adjusted provincial attack rates or with mean age of infection (Supplementary Fig. 2). Moreover, provinces in which the highest levels of mosquito abundance were predicted did not consistently report the highest burden of disease (Fig. 4a, c). One reason for these inconsistencies is that AaS relies on historical climate data from 1970 to 2000 to estimate mosquito abundance. Therefore, our estimates do not capture anomalous climatological events such as the hurricane that made

landfall in the Dominican Republic in October 2012. AaS also assumes a high level of stability in land use and minimal urbanization since the 30-year period from which the estimates were made. Regardless of the true underlying reason for these discrepancies, the spatial distributions between endemic (dengue) and emerging (chikungunya and Zika) mosquito-borne virus diseases are not very well correlated.

**Clinical outcomes of dengue cases and trends in febrile illness cases are indicators of disease prevalence.** Up to this point, our findings demonstrate that emerging mosquito-borne disease outbreaks are not confined to the temporal and spatial patterns of endemic disease outbreaks in the Dominican Republic (Figs. 1–4).

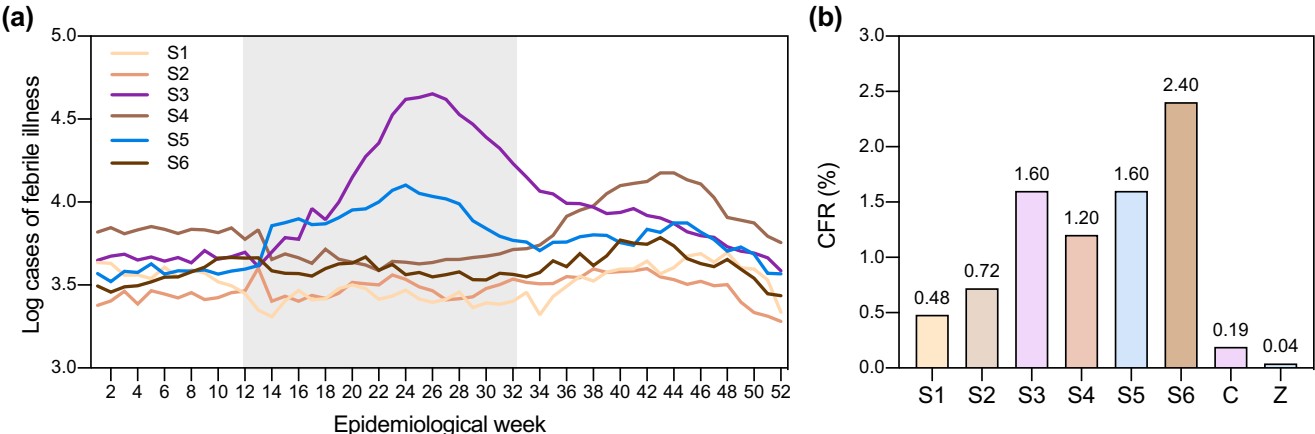

**Fig. 5 Reported febrile illness cases and case fatality rates of dengue cases during inter-dengue outbreak periods exceeded those reported during dengue outbreaks. a** Temporal trends in weekly febrile illness cases reflect temporal trends in reported dengue, chikungunya, and Zika cases. Acute febrile illness cases reported by epidemiological week per municipality were obtained from the National Statistics Directorate of the Ministry of Health (saip.gob. do). Weekly cases were aggregated to the national level. Chikungunya and Zika outbreak peaks are denoted by the gray box. **b** Percent case fatality rates (CFR) for dengue cases rose during seasons in which no dengue outbreak was reported. CFRs were calculated for six seasons of reported dengue cases (S1–S6) and two seasons of emerging disease cases (chikungunya (C) and Zika (Z)).

Although similar inconsistencies in outbreak dynamics have been observed elsewhere in the Americas (Supplementary Fig. 3)[23,24], it is difficult to discern if our findings are due to differences in epidemiology or in underlying reporting biases. Therefore, we used general patterns of febrile illness and clinical characteristics to show that reporting biases did not influence our previous conclusions (Fig. 5).

Because the vast majority of cases in our dataset reported fever independent of disease (Table 1), we hypothesized that febrile illness incidence should reflect disease incidence reported during our study period. When we examined the number of febrile illness cases reported in the Dominican Republic per epidemiological week per season, we found that temporal trends in febrile illness cases in Season 3 (2014–2015) and Season 5 (2016–2017; Fig. 5a) were consistent with those we observed in our chikungunya and Zika case data, respectively (Fig. 1a). In Season 4 (2015–2016, dengue), febrile illness cases peaked in week 44, coincident with the peak of the dengue outbreak (Fig. 1a). To ensure that the spatial relationships we identified in Fig. 4 were not purely the product of differential reporting practices, we compared population-adjusted attack rates of mosquito-borne disease and febrile illness reported by individual provinces during each season and found no significant correlation (Supplementary Fig. 4). Equally pronounced seasonal peaks of reported febrile illness were not observed in Season 1 and Season 2 despite the known dengue outbreaks that occurred within that period. We speculate that changes made in reporting protocols for febrile illness cases during the chikungunya outbreak in Season 3 prompted a sustained increase in national reporting thereafter[46].

We then investigated why reported febrile illness cases peaked seasonally in Seasons 5 and 6 despite an apparent decline in coincident dengue cases by comparing case fatality rates (CFRs) of dengue cases across seasons. Specifically, we considered the possibility that this decline was due to under-reporting of dengue cases during the interim periods between dengue outbreaks (Seasons 3, 5, and 6). The average CFR during these seasons was 1.87% (range: 1.60–2.40%), whereas the average CFR of reported cases during the three large dengue outbreaks in Seasons 1, 2, and 4 was 0.8% (range: 0.48–1.20%) (Fig. 5b). A wide range of dengue CFRs is reported in the literature, with a mean of 1.62% (95% CI: 0.64–4.02%)[47], and without serosurveillance data, we are unable to rule out the possibility that the circulation of different dengue serotypes resulted in differences in seasonal CFRs. However, if we assume that fatality among cases during the third dengue outbreak (Season 4) were well reported, dengue cases were under-reported by 33% in Season 3 and Season 5, and by twofold in Season 6. These are likely conservative estimates as the current surveillance system is passive and therefore does not capture asymptomatic cases. As a result, Seasons 3, 5, and 6 may represent periods characterized by a large number of mild cases that were not captured by the surveillance system. It is also likely that the number of deaths owing to chikungunya was substantially under-reported; however, our calculation of 0.19% corresponds to a corrected, post hoc estimate of this rate (0.15%) for the Dominican Republic[48].

## Discussion

Our study demonstrates that, even when transmitted by the same mosquito vector, viruses are not beholden to the same temporal and spatial outbreak dynamics. Instead, when and where the new virus is introduced, the size of the susceptible human population, and the capacity of local surveillance systems determine these dynamics. In short, dengue epidemiology cannot be used to anticipate the location and timing of future emerging mosquito-borne disease outbreaks in the Dominican Republic, and likely in other Caribbean countries and territories. Instead, consistent and sustainable surveillance methods should be implemented to limit disease and prevent future outbreaks. These methods could include serosurveillance of the population during periods between outbreaks[49], testing local mosquito vectors for viral infections[50–52], and monitoring health outcomes of travelers who visit the country[53].

Maintaining a sustainable surveillance system is critical for preventing the silent transmission of viruses that can fuel large outbreaks. Other countries in the Americas reported subsequent outbreaks of chikungunya and Zika after their initial outbreaks. We cannot conclusively determine whether the Dominican Republic experienced a similar pattern because surveillance data for chikungunya and Zika are not available for seasons following the initial outbreaks of these diseases. Elucidating whether dengue, chikungunya, and Zika are co-circulating in the country will be critical for triaging and providing appropriate clinical care to patients who present with febrile illness[54], especially if chikungunya and Zika virus transmission is now in sync with

dengue transmission[55,56]. Understanding the role of immunity in modulating the rate of arbovirus spread in the population will help to clarify this latter point. Such a relationship has been observed in the context of vaccination campaigns, during which annual viral outbreak peaks shift later in the year as the population is immunized[57]. For this reason, the frequency of outbreaks across years likely does impact the timing of the individual outbreaks and may cause arboviral outbreaks to become synced.

Equally important is the identification of viruses that could spread across international borders. In our study, we observed an epidemiological link between Zika outbreaks in Haiti and the southwest region of the Dominican Republic. This connection likely extends to other mosquito-borne viruses such as Mayaro virus and West Nile virus, which have been reported in Haiti[58–60]. The former can be transmitted by *Aedes* mosquitoes[14,61–63], and the latter has been isolated from zoonotic reservoirs in the Dominican Republic[64]. Similarly, reports of international travelers carrying mosquito-borne viruses from the Dominican Republic back to their home countries demonstrate that gaps in surveillance have global implications[65–70]. Therefore, understanding the transmission patterns of viruses and developing a unified, international plan to combat them before they cause an outbreak will help mitigate the potential for such an event.

Our findings suggest that dengue cases were under-reported following the Zika outbreak in 2016; however, there are a number of possible explanations for the ostensible decline in dengue cases that should be explored. Widespread mosquito control measures motivated by the Zika outbreak could have limited the spread of dengue later that year. Although plausible, this line of reasoning does not explain why a similar post-Zika decline was observed in other countries in the Americas[71], nor why there was a resurgence of dengue cases in 2019. A second explanation is that Zika infections confer some level of temporary immunity to subsequent dengue infections[72–75]. This theory cannot account for the small number of dengue cases reported in 2014 following the chikungunya outbreak, as the etiological agent of that disease is an alphavirus, and it assumes very high attack rates and extensive under-reporting of Zika in 2016 to have achieved sufficient levels of herd immunity. Our findings instead suggest that prior Zika infection protects against symptomatic dengue infections because we observed significant positive correlation between dengue attack rates within provinces across outbreaks. Given that cross-reacting immunity between dengue serotypes is well documented[76,77], our data suggest that a similar relationship between dengue and Zika would not result in a widespread decline of cases. Rather, dengue transmission could have reasonably persisted undetected if most of those infected were not hospitalized. If true, this hypothesis would explain why seasonal peaks in reported febrile illness cases persisted in 2017 and 2018, and why CFRs among reported dengue cases appeared to be elevated in Seasons 5 and 6 if the true number of cases was under-reported. To better understand these complex interactions, the collection of serotype information should be incorporated into current dengue surveillance efforts.

There are a few important limitations to our study. First, our dataset included chikungunya and Zika case data from the initial wave of each disease, and we cannot therefore compare temporal and spatial dynamics of these diseases across seasons. After these initial outbreaks, diagnostic testing for these diseases has largely ceased. Although the number of cases of these diseases reported in the Dominican Republic has declined to zero, the true burden of disease is unknown. Future studies should investigate whether these viruses have continued to circulate undetected in the country and whether their spatiotemporal dynamics have since synchronized with that of dengue virus. Second, the reporting system for suspected chikungunya cases differed from that used

for suspected Zika cases. During the chikungunya outbreak, most febrile illness cases without apparent cause were initially classified as suspected chikungunya cases. For this reason, the number of cases reported by the Pan-American Health Organization (PAHO) and the Ministry of Health was significantly larger than those, which we have reported here[78]. Our chikungunya case data contains a disproportionate number of children in the <1 year age group, indicating that the dissemination of diagnostic testing may have been skewed toward high-risk groups. Third, our findings demonstrate that an epidemiological relationship existed between the Dominican Republic and Haiti during the Zika epidemic in 2016, but we cannot determine the directionality of cross-border virus movement without virus genomic data. However, given that mosquitoes do not recognize political boundaries, it is possible that the infected vectors themselves move between countries. More likely, human movement between the two countries facilitated by the main roadway (Fig. 1e) drives the longer-distance, international spread of the viruses[79]. Regardless of the exact mechanism, it can be assumed that bi-directional spillover of mosquito-borne diseases will occur in the future unless appropriate binational surveillance and control measures are implemented. Finally, our analysis primarily focused on virus transmission by *Aedes aegypti* mosquitoes, but it is possible that other mosquito vectors contributed to the propagation of the outbreaks we investigated. Specifically, *Aedes albopictus* may have played a key role in chikungunya transmission[80,81]. Limited data are available on the distribution of relevant mosquito vectors in the country[82,83], and broader entomological surveillance is needed to better address this question.

Taken together, our study demonstrates that surveillance for mosquito-borne diseases should be sustained during periods when transmission appears to be low because patterns in reported dengue cases are poor indicators of future emerging mosquito-borne virus outbreak dynamics. Reported symptoms and case demographics may be useful for identifying shifts in disease prevalence, but many clinical features, especially fever, that we have noted are likely a function of the reporting and diagnostic algorithms used during an outbreak. Active reporting of new dengue, chikungunya, and Zika cases and the broader deployment of diagnostics for newly emerged diseases are needed to ascertain more accurate case profiles. Outbreaks of emerging tropical diseases are a threat to the public health of the Caribbean, and endemic diseases such as dengue precipitate health crises with increasing frequency. Given the pervasiveness of mosquito-borne diseases in tropical climates, sustainable surveillance systems rather than reactionary disease control measures should be implemented to prevent future crises.

## Methods

**Description of data**. Data for suspected and confirmed cases of dengue, chikungunya, and Zika reported between 2012–2018 were extracted from the Dominican Republic Ministry of Health Weekly Reports (digepisalud.gob.do). Depersonalized demographic and clinical characteristics of cases were solicited from the National Statistics Directorate of the Ministry of Health (MoH) (saip.gob.do). Data were organized by date of onset of symptom report.

The National Epidemiology Directorate of the MoH collects dengue case reports by passive surveillance in a weekly manner in its Epi 1 form within their Sistema Nacional de Vigilancia Epidemiológica (SINAVE) Digital Platform for every healthcare setting across the country. These data were collected in weekly reports submitted to their website (digepi.gob.do) and organized by province and week of reported cases. The data collected by the MoH includes age, sex, province, and municipality of residence, date of symptom onset, clinical outcome, and symptomatology. The MoH included mandatory reports from suspected and confirmed chikungunya infection through 2014 and Zika virus throughout 2016, without continuing to do so thereafter.

Province- and municipality-level weekly acute febrile illness data from 2012 to 2018 were solicited from the National Epidemiology Directorate of the Ministry of Health (saip.gob.do), which collects the data as part of a passive surveillance

system. In 2014, the Ministry of Health increased its efforts to identify febrile illness cases[46].

Because the case data described above are publicly available upon solicitation to the proper institutions and consist of depersonalized data, we did not need to seek ethical approval per our IRB.

Daily climatological variables for mean temperature and relative humidity for five cities (Santo Domingo, Distrito Nacional; Barahona, Barahona; La Union, Puerto Plata; Cabrera, María Trinidad Sánchez; Santiago, Santiago) was collected from the National Meteorology Office (ONAMET) database (saip.gob.do) from January 2012 through December 2018. Hourly climatological data for Bonao, Monseñor Nouel for the same timeframe were obtained from openweathermap. org.

Population data were extracted from the 9th National Population and Household Census. This census was conducted in 2010 by the National Statistics Office[84].

Reported cases of dengue, chikungunya, and Zika for Haiti, Bolivia, Jamaica, and Venezuela were extracted from the Pan-American Health Organization website and PLISA database[85]. Dengue and Zika cases, which were reported by epidemiological week, were aggregated by month to allow for a direct comparison to chikungunya cases, which were reported by month.

**Time to outbreak peak**. Using the weekly reported data collected by the MoH, we calculated the time to peak for provincial outbreaks by first identifying the epidemiological week in which the first case of the national outbreak was reported (chikungunya: EW 6, Zika: EW 1, dengue (3): EW 14) and then counting the number of weeks elapsed until each of the 32 provinces reported the maximum number of cases for the corresponding outbreak. The mean time to peak for each outbreak was compared using a one-way analysis of variance (ANOVA) test with Tukey's multiple comparison test implemented in Prism v8.4.2.

**Transmission potential**. We calculated weekly transmission potential (*Index P*) with the Bayesian approach developed by Obolski et al.[17]. In brief, we extracted daily average temperature and relative humidity for six cities from the National Meteorology Office database or, in the case of Monseñor Nouel province, openweathermap.org. Rarely, temperature or humidity were not available for a given day. In these cases, we averaged the respective variable from the same date across the remaining 6 years. We did this for 14 days for Barahona and Puerto Plata, 3 days for Santiago, and 1 day for María Trinidad Sánchez.

We then used the R package MVSE and entomological and epidemiological priors documented in the literature (Supplementary Table 1) to calculate daily transmission potential, from which we calculated weekly means for the duration of our study period[17]. We found that the model was reasonably robust to a range of priors for each of the parameters and therefore elected to use short human incubation and infectious period estimates to inform the model. We adopted our prior estimate of human life expectancy to the Dominican Republic based on estimates by the United Nations Development Program[86], but otherwise used the same priors as reported by Oboloski et al.[17] (Supplementary Table 1).

**Effective reproduction number (*R*eff) estimates**. We estimated *R*eff for the five outbreaks using the method developed by Lipsitch et al.[87] to fit a linear regression model to our case data Eq. (1).

$$R_{eff} = r^2 * (1 - f) * f * v^2 + r * v + 1 \qquad (1)$$

We used a range of values for latent and infectious periods that were well reported in the literature to calculate *f*, the proportion of the incubation period in the serial interval, and *v*, the serial interval measured in weeks (Supplementary Table 2). Specifically, we estimated the minimum *R*eff values using the minimum estimates of the incubation period and serial interval for each disease (Supplementary Table 2), to calculate *f* and *v*, respectively. We repeated this process with the maximum estimates for each interval to obtain the maximum values for *R*eff. The median of each pair of values is indicated by horizontal bars in Fig. 3a. We obtained *r*, the epidemic growth rate, by fitting a generalized linear model to the cumulative reported cases for each outbreak and extracting the slope from the model. All calculations were done using Rv4.0.0.

**Adjusted attack rates and linear regression**. Province-level attack rates by outbreak were age- and sex-adjusted using the direct standardization method, with daily reported cases as the input and the national population as the reference population.

We calculated Pearson's R correlation coefficient for province-level attack rates between various pairs of outbreaks (Fig. 4a, b). An outlier analysis showed that the removal of outliers did not substantially affect the size or significance of the correlation coefficients, so we included all data points.

**Aedes aegypti suitability score (AaS)**. We used AaS values calculated for each month by Kraemer et al.[18], who used both climate and land-use data on a 5 km × 5 km scale collected from 1970 and 2000 and compiled by WorldClim. Monthly suitability scores were extracted and averaged by province in Rv4.0.0.

**Reporting summary**. Further information on research design is available in the Nature Research Reporting Summary linked to this article.

## Data availability

With the exception of global AaS and surveillance data, source data are available with this paper (https://github.com/grubaughlab/Paper_arbovirus_Epi_DR)[88]. Surveillance data are available upon solicitation of the Dominican Republic Ministry of Health SAIP (Solicitud de Acceso a Informacion Publica/Solicitation of Publicly Available Information; website: saip.gob.do). Climatological data may be obtained from the National Meteorology Office (ONAMET). Depersonalized demographic and clinical characteristics of cases may be obtained from the National Statistics Directorate of the Ministry of Health (MoH). Weekly dengue, chikungunya, and Zika case data may be obtained from the Sistema Nacional de Vigilancia Epidemiológica (SINAVE). Febrile illness case data may be obtained from the National Epidemiology Directorate of the Ministry of Health. Global AaS values may be available upon request. Source data are provided with this paper.

## Code availability

The R scripts for this manuscript are available on GitHub (https://github.com/grubaughlab/Paper_arbovirus_Epi_DR).

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

## Acknowledgements
We thank C. Vogels, A. Brito, J. Fauver, C. Kalinich, I. Ott, S. Lapidus, K. Gangavarapu, J. Pack, and S. Taylor for feedback and/or assistance with the methodology. The study was funded in part by the Hecht Global Health Faculty Network Award provided to N.D. G. from the Yale Institute of Global Health.

## Author contributions
M.E.P., L.T., and N.D.G. conceived of the study. M.E.P. and R.E. designed the analyses. M.E.P., R.E., J.L., M.U.G.K., R.P., N.D.G., and L.T. all contributed to data analysis and writing the paper.

## Competing interests
The authors declare no competing interests.
