## [Peer Review File · Nature Communications]

REVIEWER COMMENTS

Reviewer #1 (Remarks to the Author):

The manuscript evaluates the asynchronicity of arboviruses outbreaks in the Dominican Republic. The authors use suspect case data of dengue, Zika, and chikungunya during the period 2012 and 2018. They have shown that outbreaks of the emerging arbovirus (ZIKV and CHIKV) occur in a different period the seasonal dengue outbreaks. They also show that Zika and Chikungunya outbreaks occur in sub-optimal climate conditions which emphasizes the importance of the size of the susceptible human population. The manuscript is well written and I have two main questions for the authors:

- Lines 138-139: The authors hypothesized that early Zika virus transmission could have occurred in the south-western region because of the region's proximity to Haiti. Perhaps this could be checked by analyzing the time series of cases reporting rash as a proxy of a Zika case. If the hypothesis is correct we would observe several rash reports in dengue and chikungunya cases greater than expected for these diseases. Also, the microcephaly cases reported in early 2016 (<https://www.ncbi.nlm.nih.gov/pmc/articles/PMC6346438/>) support the hypothesis that Zika was already circulating before 2016 in the country. Could this analysis be done with the available dataset?

- Lines 236-237: The authors mention the *Aedes aegypti* suitability score, presented in Fig 4c and Fig. S1. It is clear there is some spatial heterogeneity of the mosquito abundance (AsS) but there is no evidence that the range of the AsS values is large enough to make the regions different, looking at the Fig S1 values varies from -1.6 to -1.1. Looking at the extreme values, are the mosquito abundance in regions with AsS around -1.5 too different than regions with larger values, i.e. around -1.1? Could the AsS score be clarified?

Reviewer #2 (Remarks to the Author):

The authors investigate the timing of outbreaks of various mosquito-borne viral diseases, comparing endemic dengue outbreaks to emerging Chikungunya and Zika outbreaks. The timing of endemic outbreaks did not predict timings of epidemic outbreaks, which could not be explained by differences in climatic factors. The work presented in this manuscript offers interesting descriptive results regarding differences in disease dynamics, even when spread by the same vector and within the same country.

Overall, the methods section would benefit from some elaboration so that readers can better understand how conclusions were drawn. This would also aid reproducibility.

Specific comments below:

RESULTS

Figure 1e is difficult to read because the line representing the road connecting Haiti and the Dominican Republic appears similar to the line for the border of Haiti. Differentiating either using a different color or different pattern would help interpretation

In comparing outbreak dynamics to climatic factors using Index P in lines 166-188, more information is needed regarding how the conclusions were drawn, either here or in the methods section. Was a specific test or analysis conducted, or was this based on visual examination?

The statement on lines 199-200 states the conclusions from the previous section very clearly. Including such a statement in the previous section would help present the overall findings from

that section.

The statement on lines 243-247 describes explanations for results seen. This may be more suitable in the discussion section.

DISCUSSION

Lines 368-371: It would help to elaborate if the movement of viruses between the countries would be driven more by mosquito movement or human movement (if this is known/hypothesized).

Clear differences are shown across different diseases. Are there any indications, whether through literature or the analyses in this study, of whether Chikungunya and Zika show different patterns because they are different viruses or because they are emerging? For example, are the differences in onset to peak believed to be based on characteristics of the viruses themselves or on the fact that they are seen with varying frequency across years?

METHODS

In data descriptions, it would be helpful to mention which analyses/results each data source was used for.

Line 435: How were data aggregated? Weekly mean?

Line 443-451: The setup of the linear regression is unclear. Equation 1 and its description seem as if values of r , f , and v are derived from literature (Table S2) and then inserted into an equation for R_{eff} . A short description of the type of regression model and data used to inform it would be beneficial for reproducibility.

Reviewer #1

The manuscript evaluates the asynchronicity of arboviruses outbreaks in the Dominican Republic. The authors use suspect case data of dengue, Zika, and chikungunya during the period 2012 and 2018. They have shown that outbreaks of the emerging arbovirus (ZIKV and CHIKV) occur in a different period the seasonal dengue outbreaks. They also show that Zika and Chikungunya outbreaks occur in sub-optimal climate conditions which emphasizes the importance of the size of the susceptible human population. The manuscript is well written and I have two main questions for the authors:

We greatly appreciate the positive feedback provided by the reviewer, and we have addressed all of their concerns below.

- Lines 138-139: The authors hypothesized that early Zika virus transmission could have occurred in the south-western region because of the region's proximity to Haiti. Perhaps this could be checked by analyzing the time series of cases reporting rash as a proxy of a Zika case. If the hypothesis is correct we would observe several rash reports in dengue and chikungunya cases greater than expected for these diseases. Also, the microcephaly cases reported in early 2016 (<https://www.ncbi.nlm.nih.gov/pmc/articles/PMC6346438/>) support the hypothesis that Zika was already circulating before 2016 in the country. Could this analysis be done with the available dataset?

We thank the reviewer for this interesting suggestion. **To investigate the incidence of rash among dengue cases in different regions of the Dominican Republic, we calculated the proportion of dengue cases reporting a rash over time by region.** We observed that Enriquillo, the south-western region that we hypothesized had early Zika virus transmission, reported a higher monthly proportion of dengue cases with rash compared to other regions throughout much of 2015, with a sharp increase in early 2016. This provides further support for the hypothesis of early Zika virus transmission in Enriquillo. We did not conduct the same analysis using chikungunya cases since we only had one year of data.

We added the following statement to Lines 143-145 of the text:

“Moreover, the proportion of dengue cases reporting a rash was larger and peaked earlier in the south-western region Enriquillo in 2016 compared to the rest of the country (Fig. S1). This suggests that some Zika cases may have been misclassified as dengue at the beginning of the outbreak.”

We added a supplementary figure (Fig. S1) to illustrate these findings:

Figure S1: The proportion of dengue cases who reported experiencing a rash peaked earlier in the south-western region Enriquillo compared to the rest of the country. The number of dengue cases with rash were summed for each month. The gray box denotes when the Zika outbreak occurred.

- Lines 236-237: The authors mention the *Aedes aegypti* suitability score, presented in Fig 4c and Fig. S1. It is clear there is some spatial heterogeneity of the mosquito abundance (AsS) but there is no evidence that the range of the AsS values is large enough to make the regions different, looking at the Fig S1 values varies from -1.6 to -1.1. Looking at the extreme values, are the mosquito abundance in regions with AsS around -1.5 too different than regions with larger values, i.e. around -1.1? Could the AsS score be clarified?

The *Aedes aegypti* suitability score (AaS), which represents the probability of the presence of the *Aedes aegypti* mosquito vector, is bounded between 0 and 1 when normalized by its maximum value. **We modified the Fig. 4c color scheme to include equally-sized bins to better illustrate the observed spatial heterogeneity. In Fig. S1, we changed the AaS scale to be bounded between 0 and 1 (versus using a log₁₀ scale) to improve interpretability.**

Reviewer #2:

The authors investigate the timing of outbreaks of various mosquito-borne viral diseases, comparing endemic dengue outbreaks to emerging Chikungunya and Zika outbreaks. The timing of endemic outbreaks did not predict timings of epidemic outbreaks, which could not be explained by differences in climatic factors. The work presented in this manuscript offers interesting descriptive results regarding differences in disease dynamics, even when spread by the same vector and within the same country. Overall, the methods section would benefit from some elaboration so that readers can better understand how conclusions were drawn. This would also aid reproducibility. Specific comments below:

We thank the reviewer for their feedback, and we address their concerns below.

RESULTS

Figure 1e is difficult to read because the line representing the road connecting Haiti and the Dominican Republic appears similar to the line for the border of Haiti. Differentiating either using a different color or different pattern would help interpretation

We agree with the reviewer on this observation. **We have changed the color of the road to blue.**

In comparing outbreak dynamics to climatic factors using Index P in lines 166-188, more information is needed regarding how the conclusions were drawn, either here or in the methods section. Was a specific test or analysis conducted, or was this based on visual examination?

We apologize for not being clearer on this point. Due to the temporal limitations of our data for the chikungunya and Zika outbreaks (one year), we were not able to justifiably conduct a statistical test. **To clarify this, we have edited the following statements:**

Lines 177-180:

“When we compared the temporal dynamics of transmission potential to reported disease incidence, we found that the number of emerging disease cases reported weekly peaked before transmission potential had reached a seasonal maximum for both outbreaks (chikungunya and Zika), while the number of dengue cases reported weekly peaked after this point for **two of the three** dengue outbreaks (Fig. 2a). **We measured the number of weeks elapsed between outbreak cases and *Index P* during each season and found that, on average, dengue cases peaked 17 weeks after *Index P*. In contrast, chikungunya cases peaked 14 weeks and Zika cases peaked 27 weeks earlier than *Index P* (Fig. 2a, circles).”**

We added circles to Fig. 2a to supplement this point and edited the caption accordingly (Line 191).

Lines 184-185:

“Despite this variation, the timing of provincial outbreaks conformed to the trend observed **with visual examination** on the national level:”

The statement on lines 199-200 states the conclusions from the previous section very clearly. Including such a statement in the previous section would help present the overall findings from that section.

We thank the reviewer for this suggestion and **have rephrased Lines 165-166 as follows:**

“**Having observed that the emerging disease outbreaks peaked earlier in the year than the three dengue outbreaks (Fig. 1), we considered two possible explanations.**”

The statement on lines 243-247 describes explanations for results seen. This may be more suitable in the discussion section.

We appreciate the reviewer’s feedback on this point; however, we have elected to keep this statement in the results section so that the reader does not come away from this section with the false impression that land use and climate do not influence mosquito behavior and transmission. Rather our results likely reflect the limitations of the method we used.

DISCUSSION

Lines 368-371: It would help to elaborate if the movement of viruses between the countries would be driven more by mosquito movement or human movement (if this is known/hypothesized).

We apologize for not making this point more thoroughly. **To elaborate, we have modified Lines 368-371 as follows:**

“Third, our findings demonstrate that an epidemiological relationship existed between the Dominican Republic and Haiti during the Zika epidemic in 2016, but we cannot determine the directionality of cross-border virus movement without virus genomic data. **However, given that mosquitoes do not recognize political boundaries, it is possible that the infected vectors themselves move between countries. More likely, human movement between the two countries facilitated by the main roadway (Fig. 1e) drives the longer-distance, international spread of the viruses (Verdonschot & Besse-Lototskaya, 2014). Regardless of the exact mechanism,** it can be assumed that bi-directional spillover of mosquito-borne diseases will occur in the future unless appropriate bi-national surveillance and control measures are implemented.”

Verdonschot, Piet F. M., and Anna A. Besse-Lototskaya. 2014. “Flight Distance of Mosquitoes (Culicidae): A Metadata Analysis to Support the Management of Barrier Zones around Rewetted and Newly Constructed Wetlands.” *Limnologica* 45 (March): 69–79.

Clear differences are shown across different diseases. Are there any indications, whether through literature or the analyses in this study, of whether Chikungunya and Zika show different patterns because they are different viruses or because they are emerging? For example, are the differences in onset to peak believed to be based on characteristics of the viruses themselves or on the fact that they are seen with varying frequency across years?

We thank the reviewer for raising this important point. We believe the temporal differences are attributable to the fact that chikungunya and Zika were emerging viruses during the time period captured in our study. Specifically, we think these differences are due to the level of population immunity, which likely modulates the speed at which viruses can spread through a population. This phenomenon has been observed in the context of vaccination campaigns. Specifically, annual viral outbreak peaks are observed to shift later in the year as the population is immunized (Shioda, *et al.* Identifying signatures of the impact of rotavirus vaccines on hospitalizations using sentinel surveillance data from Latin American countries. *Vaccine*. 2020 Jan. 10). For this reason, as the reviewer notes, the frequency of outbreaks across years likely does impact the timing of the individual outbreaks, but this is not a consequence of an intrinsic characteristic of the viruses.

There is also evidence that chikungunya and Zika cases sync up with dengue cases in years subsequent to their emergence. Faria *et al.* describe this phenomenon in their analysis of the epidemiology of Chikungunya virus in Bahia, Brazil (Faria N, *et al.* *Epidemiology of Chikungunya Virus in Bahia, Brazil, 2014-2015. PLOS Currents Outbreaks*. 2016 Feb 1). Similarly, we observed that post-emergent Zika outbreaks coincided with dengue outbreaks elsewhere in South America (**Fig. S2**).

It is possible that the length of the incubation or infectious periods could modulate the speed at which each virus spread through the population. However, we did not find either Zika or chikungunya to be an outlier compared to dengue in this respect (**Table S2**). Similarly, because these viruses are all mainly transmitted by the same mosquito vector, there is no evidence for drastically different extrinsic incubation periods or time to transmission from vector to host that could satisfactorily explain our observations.

To address these points, we have added the following to our discussion section:

“Elucidating whether dengue, chikungunya, and Zika are co-circulating in the country will be critical for triaging and providing appropriate clinical care to patients who present with febrile illness (Vogels et al., 2019), especially if chikungunya and Zika virus transmission is now in sync with dengue transmission (Bisanzio et al., 2018; L. P. Freitas et al., 2019). **Understanding the role of immunity in modulating the rate of arbovirus spread in the population will help to clarify this latter point. Such a relationship has been observed in the context of vaccination campaigns, during which annual viral outbreak peaks shift later in the year as the population is immunized (Shioda et al., 2020). For this reason, the frequency of outbreaks across years likely does impact the timing of the individual outbreaks and may cause arboviral outbreaks to become synced.**”

METHODS

In data descriptions, it would be helpful to mention which analyses/results each data source was used for.

We apologize for not making this more explicit in our methods section. **To clarify, we have added the following statements:**

Lines 423-424

“**Using the weekly reported data collected by the MoH**, we calculated the time to peak for provincial outbreaks by first identifying the epidemiological week in which the first case of the national outbreak was reported...”

Lines 430-431

“Briefly, we extracted daily average temperature and relative humidity for 6 cities **from the National Meteorology Office database or, in the case of Monseñor Nouel province, openweathermap.org.**”

Lines 454-455

“We calculated province-level attack rates by outbreak that were age- and sex-adjusted using the direct standardization method, with **daily reported cases as the input and** the national population as the reference population.”

Line 435: How were data aggregated? Weekly mean?

We calculated the mean *Index P* for each week. **We have updated Line 435 as follows:**

“We then used the R package MVSE and entomological and epidemiological priors documented in the literature (**Table S1**) to calculate daily transmission potential, **from which we calculated weekly means for the duration of our study period.**”

Line 443-451: The setup of the linear regression is unclear. Equation 1 and its description seem as if values of r , f , and v are derived from literature (Table S2) and then inserted into an equation for R_{eff} . A short description of the type of regression model and data used to inform it would be beneficial for reproducibility.

We thank the reviewer for this comment as it led us to identify a minor error in the values we used to calculate R_{eff} . The values and R_{eff} estimates have been updated accordingly in Fig. 3 and Table S2. These changes did not affect the qualitative result of our analysis or any of the conclusions we drew from it.

We have added a more thorough explanation to this part of the methods section as follows:

We used a range of values for latent and infectious periods that were well reported in the literature to calculate f , **the proportion of the incubation period in the serial interval**, and v , **the serial interval measured in weeks**. **Specifically, we estimated the minimum R_{eff} values using the minimum estimates of the incubation period and serial interval for each disease (Table S2), to calculate f and v , respectively. We repeated this process with the maximum estimates for each interval to obtain the maximum values for R_{eff} . The median of each pair of values is indicated by horizontal bars in Fig. 3a.**

We obtained r , the epidemic growth rate, **by fitting a generalized linear model to the cumulative reported cases for each outbreak and extracting the slope from the model.**

REVIEWERS' COMMENTS

Reviewer #1 (Remarks to the Author):

The authors have properly addressed my questions. They provide good evidence for the asynchronicity of Zika and chikungunya outbreaks in Dominican Republic. Their analysis suggests we shall observe asynchronicity in other regions and countries where dengue is endemic, as an example some regions in Brazil.

Leo Bastos (Oswaldo Cruz Foundation)

Reviewer #2 (Remarks to the Author):

The previous comments were addressed well. The manuscript is suitable for publication.

Reviewer #1

The authors have properly addressed my questions. They provide good evidence for the asynchronicity of Zika and chikungunya outbreaks in Dominican Republic. Their analysis suggests we shall observe asynchronicity in other regions and countries where dengue is endemic, as an example some regions in Brazil. Leo Bastos (Oswaldo Cruz Foundation)

We thank Dr. Bastos for the positive feedback.

Reviewer #2

The previous comments were addressed well. The manuscript is suitable for publication.

We thank the reviewer for their recommendation.